# Rheological Study of Phenol Formaldehyde Resole Resin Synthesized for Laminate Application

**DOI:** 10.3390/ma13112578

**Published:** 2020-06-05

**Authors:** Nuruldiyanah Kamarudin, Dayang Radiah Awang Biak, Zurina Zainal Abidin, Francisco Cardona, Salit Mohd Sapuan

**Affiliations:** 1Department of Chemical and Environmental Engineering, Faculty of Engineering, Universiti Putra Malaysia, Serdang 43400 UPM, Selangor, Malaysia; nuruldiyanahkamarudin@gmail.com (N.K.); zurina@upm.edu.my (Z.Z.A.); 2Institute of Advance Technology, Universiti Putra Malaysia, Serdang 43400 UPM, Selangor, Malaysia; 3Safety Engineering Interest Group, Universiti Putra Malaysia, Serdang 43400 UPM, Selangor, Malaysia; 4Aerospace Manufacturing Research Centre (AMRC), Faculty of Engineering, Universiti Putra Malaysia, Serdang 43400 UPM, Selangor, Malaysia; cardonafrancisco2003@yahoo.com.au; 5Advanced Engineering Materials and Composites Research Centre, Department of Mechanical and Manufacturing Engineering, Universiti Putra Malaysia, Serdang 43400 UPM, Selangor, Malaysia; sapuan@upm.edu.my

**Keywords:** phenol formaldehyde resin, paraformaldehyde, laminate resin, fluid behavior, rheology

## Abstract

Heat explosions are sometimes observed during the synthesis of phenol formaldehyde (PF) resin. This scenario can be attributed to the high latent heat that was released and not dissipated leading to the occurrence of a runaway reaction. The synthesis temperature and time played important roles in controlling the heat release, hence preventing the resin from hardening during the synthesis process. This study aims to assess the rheological and viscoelasticity behaviors of the PF resin prepared using paraformaldehyde. The prepared PF resin was designed for laminate applications. The rheological behavior of the PF resin was assessed based on the different molar ratios of phenol to paraformaldehyde (P:F) mixed in the formulation. The molar ratios were set at 1.00:1.25, 1.00:1.50 and 1.00:1.75 of P to F, respectively. The rheological study was focused at specific synthesis temperatures, namely 40, 60, 80 and 100 °C. The synthesis time was observed for 240 min; changes in physical structure and viscosity of the PF resins were noted. It was observed that the viscosity values of the PF resins prepared were directly proportional to the synthesis temperature and the formaldehyde content. The PF resin also exhibited shear thickening behavior for all samples synthesized at 60 °C and above. For all PF resin samples synthesized at 60 °C and above, their viscoelasticity results indicated that the storage modulus (G′), loss modulus(G″) and tan δ are proportionally dependent on both the synthesis temperature and the formaldehyde content. Heat explosions were observed during the synthesis of PF resin at the synthesis temperature of 100 °C. This scenario can lead to possible runaway reaction which can also compromise the safety of the operators.

## 1. Introduction

Phenol formaldehyde (PF) resin-based materials have been widely used as adhesives, composite materials, molds and laminates [1,2,3,4,5,6,7,8]. PF resin is obtained from the condensation reaction between phenol and formaldehyde in the presence of an alkaline catalyst [8,9,10,11,12,13,14]. It is commonly synthesized by reacting liquid formaldehyde (formalin) with phenol in the presence of a catalyst [10,14,15,16,17,18,19]. Pilato [20] and Gardziella et al. [12] described that liquid formaldehyde has higher chemical reactivity to form methylol phenols compared to paraformaldehyde because of the availability of free formaldehyde molecules. 

However, various studies [15,18,19] had showed that when formalin was used to synthesize PF resin, a much longer synthesis period was required for the reaction to complete. The amount of water in formalin is about 70% by weight. During the PF resin synthesis, more water is generated via the condensation reaction and thus increases the amount of water in the mixture. Water removal is critical to make sure the resin is viscous with high (approximately 75%–80% by weight) solid content [21,22,23,24]. Some studies had shown that fast removal of water, i.e., via evaporation at high temperature, led to the formation of voids within the PF resin structures. These voids create defects on the cured resin and might affect the structural integrity and mechanical properties of the produced laminate [8,16,20]. The recommended viscosity for synthesized PF resin that made it easily applied on fiber as laminating material is approximately 400–600 cP [21]. In some studies, different formulation required different synthesis time to achieve specific viscosity. To illustrate, Shafizadeh et al. [15] whom used formalin in the synthesis of PF resin required nine hours to produce PF resin with a viscosity of 900 cP. In another study conducted by Zhang et al. [18], they required three hours to produce PF resin with viscosity of 150 cP, whilst Lin and Lee [19] required more than three hours to produce PF resin with viscosity of 142 cP. Another group of researchers synthesized PF resin using paraformaldehyde. Cardona and Sultan [8] spent only three hours to produce PF resin with a final viscosity of 2000 cP and Cui et al. [23] required two hours to produce PF resin with a viscosity value of 900 cP. The presented studies indicated that the synthesis period required to achieve a desired viscosity depended on the formulation used. 

The synthesis of PF resin from paraformaldehyde happens in three stages. The first stage is the depolymerization process in which the long chain paraformaldehyde is broken down into single CH_2_O molecule. This process occurs when paraformaldehyde solution is heated to at least 60 °C in the presence of a base catalyst. The chemical reaction of the depolymerization of paraformaldehyde is depicted in Figure 1. 

The second stage of the PF resin synthesis is the reaction between CH_2_O molecule with phenol to form hydroxymethyl phenol. This is an addition reaction in which the double bond that binds atom O and C in the single formaldehyde molecule is broken down and attached to the phenol ring to form either ortho- and para- hydroxymethyl phenol [26]. The addition reaction is shown in Figure 2. 

The hydroxymethyl phenols then condense to form low molecular weight prepolymers. These prepolymers will transform to various molecular weight polymers with rigid cross-linked network upon heated. The rate of reaction is highly dependent on the type of catalyst used as well as the temperature of the system [26,27]. The polycondensation reaction is represented by the chemical reaction shown in Figure 3.

Phenol formaldehyde reaction is an exothermic process [20,27,28]. In this reaction heat is generated when the reaction is initiated. The generated heat will drive the reaction faster and thus generate more heat in the reactor. Accumulation of excessive heat in the reactor which cannot be properly removed by the cooling arrangement will accelerate the reaction further and cause the runaway reaction. A large increase in both temperature and pressure in the reactor may lead to an explosion [26]. Hence, it is noted that controlling and maintaining the synthesis temperature are important to maintain the required quality of the PF resin produced as well as keeping the operation safe.

Most research on PF resins focused their discussions on the synthesis routes and the application of the PF resin [15,18,19,20,23,27,28]. The study on the rheological and viscoelasticity behaviors of PF resin during the synthesis process is not available. These fluid properties are highly affected by the molar ratios of phenol and formaldehyde used, the synthesis temperature, the synthesis time as well as pH of the mixture. In this study, paraformaldehyde was used in the synthesis of PF resin. The aim of this study is to investigate the effects of molar ratios of phenol to formaldehyde used and the synthesis temperature on the viscosity of the PF resin produced. The synthesis period was observed and recorded for 240 min for all samples. The pH of the mixture was maintained.

## 2. Materials and Methods 

Phenol was purchased from Malayan Adhesive & Chemical Sdn. Bhd, (Kuala Lumpur, Malaysia) with the molecular weight of 94.11 g/mol and is 95% pure. Industrial grade paraformaldehyde (molecular weight = 30.03 g/mol, purity of 92%) was purchased from Chang Chun Plastics Co. Ltd. (Zhongshan District, Taiwan). Silicon oil (Merck KGaA, Darmstadt, Germany) was used as oil bath and used to provide heat to the system. Aqueous solution of NaOH (40% by weight) was prepared from the NaOH pellets and used as the base catalyst.

### 2.1. Synthesis of Phenol Formaldehyde (PF) Resin 

Phenol formaldehyde (PF) resin was prepared by reacting phenol (P) and formaldehyde (F) at specified molar ratios. Studies conducted by [8,17,29,30,31] were used as references for the initial settings of the molar ratios and the synthesis conditions. All operations were conducted under the fume hood. Mixing was performed at 150.00 ± 5.00 rpm using a mechanical stirrer. No cooling arrangement was provided to the system which is the limitation of the system.

### 2.2. Rheological Behavior of Phenol Formaldehyde (PF) Resin

Data from research works performed by Niederhauser and Miller [29], Shafizadeh and Seferis [30], Turunen et al. [31], Christjanson et al. [17] and Cardona and Sultan [8] were used as the initial referenced values for the molar ratios, synthesis temperatures and synthesis time. Data presented in Table 1 are the samples prepared for the rheological study. The molar ratios of phenol to formaldehyde and the synthesis temperature were varied between samples. All samples were synthesized for 240 min. Samples were removed periodically and the viscosities were measured and recorded at the specified interval. All physical changes were noted during the interval period. The relationship between shear stress and shear strain of PF resin samples produced were determined.

Some of the samples have much smaller sampling intervals because of the possible rapid changes in viscosity due to the known exothermic nature of the reaction particularly at high temperature [26]. The rapid viscosity changes might occur in some solutions due to fast reaction resulted either by the formaldehyde content or the synthesis temperature. The relationship between the dynamic viscosity of PF resin with the synthesis temperatures and molar ratios of P to F was plotted. The required standard as recommended by industry for high pressure laminate (HPL) material [21] was plotted as a horizontal dashed line at 600 cP on the viscosity plot. The HPL viscosity value is the recommended viscosity value for any pliable material that will be used as laminate. For the purpose of this work, the HPL value was used as a guide to find the feasible conditions for PF resins synthesis. 

### 2.3. Viscosity Measurement

The dynamic viscosity measurement was performed using Brookfield Viscometer Middleboro, MA, USA (DV2T). The cylindrical spindle (LV-04(64)) rotating at 200 rpm was used to measure the viscosity of the PF resin. All viscosity readings were recorded at 30 °C. The accuracy and repeatability of the Brookfield Viscometer are ±1% and 0.2% of full-scale range (FSR), respectively. The measured viscosity data for all PF resin samples were recorded and tabulated

### 2.4. Viscoelasticity Analysis

The viscoelasticity behavior of the PF resin was calculated based on the recorded time, viscosity and shear stress, τ (dynes/cm^2^) from the rheology study. The torque values were obtained from the rheometer reading performed at 30 °C. The storage modulus, G′ (Pa), loss modulus, G″ (Pa) and the damping factor, tan δ were calculated based on those described by Barnes et al. [32]. The equations used to calculate the G′, G″ and tan δ are as shown in Equations (1)–(4),
(1)$$G^{\prime }=\frac{{\eta \tau \omega }^{2}}{1+\omega^{2}\tau^{2}}$$
(2)$$G^{\prime \prime }=\frac{\eta \omega }{1+\omega^{2}\tau^{2}}$$
(3)$$tan\;\delta =\frac{G^{\prime \prime }}{G^{\prime }}$$
(4)$$\omega =\left(\frac{2\pi }{60}\right)N$$

In the above equations, ω is the angular velocity (rad/s), N is the number of revolutions per min (min^−1^) and η is the dynamic viscosity of the measured solution (cP). The graphs that display these viscoelasticity parameters were plotted and discussed.

## 3. Results and Discussions

The rheological behavior of phenol formaldehyde (PF) resin was evaluated for 240 min. The rate of viscosity changes for all samples were compared.

### 3.1. Rheological Behavior of Phenol Formaldehyde (PF) Resin

The rheological data of the phenol formaldehyde (PF) resin were tabulated based on the varied molar ratios and synthesis temperatures. The viscosity data of all samples were collected for 240 min. Apparent changes in viscosity and physical appearances of the PF resins were noted. All measured viscosity values were recorded and are presented in the graphs as shown in Figure 4. 

Shorter recorded time as shown by plots number 3, 4, 7, 8, 10, 11 and 12 indicates that the PF resin samples were already hardened after the recorded value; thus, no tangible fluid data could be measured by the equipment. The viscosity of the PF resin samples located too far below the HPL line, i.e., viscosity value is less than 300 cP, were physically very runny and cannot be applied on the fiber. The PF resins samples which have the viscosity higher than 600 cP were found to be very viscous and hence, very difficult to be applied on the fiber. Using these viscosity values and the physical observation of the products as guides, it is possible to terminate the reaction at desired time to produce pliable PF resins when both the molar ratios of P to F and the synthesis temperatures are known. Therefore, in the actual manufacturing process, the synthesis time can be tailored accordingly based on the required viscosity which defines the molecular weight of the produced resin. This process is feasible if the synthesis temperature and the molar ratio of P to F are given. 

#### 3.1.1. Influence of Synthesis Temperature on the Viscosity of PF Resin

Temperature had been known to affect the synthesis process of the thermoset polymer. In this work, the effect of temperature on the viscosity of the synthesized resin was assessed. Figure 4a–c shows the plots of dynamic viscosity of PF resin synthesized in this work. 

Figure 4a–c shows the viscosity plots of PF resins prepared at different molar ratios of P to F, in which number 1 to 12 denotes the individual viscosity plot of PF resin synthesized at different temperatures, namely 40, 60, 80 and 100 °C. Shaghaghi et al. [33] and Zhao et al. [34] described that the change in viscosity per unit time represents the reaction rate of the phenolic and epoxy resin system. The HPL line shows the reference viscosity for laminates material as that recommended by industry [21]. 

The viscosity plots provide some information on the rate of reaction of the process. Based on the slope of the plots, it can be deduced that the rate of reaction of PF resins synthesized at 40 °C was the slowest; the heat generated by the process was presumably small that no significant changes on the viscosity value of the PF resin were observed up to 240 min of the synthesis period. The viscosity values of all PF resin samples synthesized at 40 °C were less than 200 cP. The resin was quite thin making it not suitable to be used as laminating material. This can be caused by the large amount of water that was remained in the final product. The amount of heat supplied by heating the system at 40 °C was not sufficient to totally remove the water that was present in the system. Thus, synthesizing the PF resin at a very low temperature seems to be not productive and use a lot of energy. 

The viscosity of PF resin synthesized at 60 °C reached ~850 cP at the end of 240 min. It took about 160 min for the reaction rate to significantly increase; one of the factors that might trigger the reaction rate to increase was the heat generated by the exothermic reaction in the system. Theoretically, the relative amount of water that can be removed at 60 °C is small, however when the reaction proceeds, more heat was generated resulting in an increase in temperature and removal of water via evaporation. 

As seen in Figure 4, all PF resin samples prepared at 80 and 100 °C show drastic viscosity changes throughout the synthesis period of 240 min. To illustrate, viscosity of PF resins sample that is displayed as plot number 4 had a viscosity change from 50 cP at 5 min to 3000 cP at 20 min. The rate of viscosity change of this sample was quite fast, which is about 196 cP/min. When the synthesis period was prolonged further, the sample was cured before the end of the designed synthesis time, i.e., 240 min, and no viscosity data were recorded after 20 min. The steep slope of the plot for all PF resin synthesized at 100 °C shows that the high synthesis temperature increased the reaction rate and generated excessive heat to the system. The heat generated remained in the system due to the absence of cooling arrangement and has led to the formation of hardened product. The physical image of the defective PF resin synthesized at 100 °C is as shown in Figure 5. Highly exothermic reaction with large amount of water generated during the polycondensation reaction caused the formation of bubbles or foams on the surface of the solidified PF resin. 

Formation of foam or bubble during the resin synthesis created void in the internal structure of the PF resin laminates and thus weaken the overall structure. 

Observing the profiles of the plots presented in Figure 4a–c, it can be inferred that the suitable temperatures to synthesize PF resins with the given molar ratios of P to F are 60 and 80 °C. This can be further verified by the shear stress vs shear strain data shown in Figure 6 in which the PF resins synthesized at these temperatures behaved as shear thickening liquid. The PF resin samples synthesized at 100 °C also portrayed shear thickening behavior; however, the high reaction rate as described earlier made the process difficult to control leading to formation of defective products. The PF resin samples synthesized at 40 °C showed shear thinning properties which are confirmed by the low viscosity values. When applied on the fiber ply, samples synthesized at 40 °C slipped off the surface of the fiber and did not form a good laminate. Kordani and Vanini [35] and Gürgen [36] also mentioned that shear thickening property is essential in handling the resin and to prevent the separation of resin from the fiber in laminate application. Based on these observations, we had tailored the suitable processing period that met the goal of the study, i.e., producing PF resin with a viscosity value between 400–600 cP. The selected synthesis period will assist in the production of the PF resin for this work. Pilato [20] and Cardona and Sultan [8] also described that it is more favorable to use different synthesis temperature to synthesize PF resin due to the chemical reactions that will happen and the chemical bonding that can be formed at different synthesis temperature. This can be attributed to the number of prepolymers [formaldehyde monomers] formed during the depolymerization process followed by the attachment of these monomers to phenol and the cross-linking of the hydroxymethyl phenol to form rigid network resin structures. Synthesizing PF resin at low temperature requires a longer time to obtain the right product because the system has to accumulate sufficient heat to 1) break the covalent bond of the large polymer, and 2) overcome the activation energy required to initiate addition reaction. If the heat accumulated is not sufficient, then the formaldehyde monomers produced in stage 1 of PF resin synthesis as shown in Figure 1 will only react with phenol to form hydroxymethyl phenol; polycondensation reaction will occur but at a very slow rate. More heat will be required to drive the process to reach the polycondensation stage since this stage involved both the removal of water as well as cross-linking of the prepolymers. If the heat supplied is not sufficient to overcome the activation energy required for initiation of polycondensation reaction, then the cross-linking network might not happen completely resulting in thin solution of linear phenolic resin prepolymer with low molecular weight [27]. This is shown by the shear thinning behavior of the PF resin produced at 40 °C as displayed in Figure 6a, Figure 7a and Figure 8a, respectively. All other PF resin samples produced at 60, 80 and 100 °C showed shear thickening behavior as shown by Figure 6, Figure 7 and Figure 8b–d.

#### 3.1.2. Influence of Formaldehyde Molar Ratio

The effects of formaldehyde molar content on the properties and the molecular structure of PF resin are well understood. High formaldehyde molar content will improve the curing degree of the phenol formaldehyde resins and form a more complex network structure of the resin [21,26]. At lower formaldehyde molar content, the formaldehyde and phenol molecules will be predominantly at the addition stage to produce hydroxymethyl phenol resulting in a low viscosity resin [26]. Parameswaran [37] mentioned that increasing the formaldehyde molar ratio will shorten the gel time or time required by the resin to cure. It is more favorable to supply formaldehyde in excess in the reaction to ensure complete reaction of phenol [8] because formaldehyde monomer will attach at either ortho- or para- position to one benzene ring. In this work, the molar ratio of phenol to formaldehyde were varied at 1.00:1.25 (PFI), 1.00:1.50 (PFII) and 1.00:1.75 (PFIII), respectively. 

All PF resins synthesized at 40 °C showed shear thinning behaviors as that presented in Figure 6. Despite of the availability of excess formaldehyde that can hypothetically form hydroxymethyl phenol monomers and their subsequent cross-linking polymers, the reaction at 40 °C was slowed down because the heat that was supplied externally was insufficient to overcome the activation energy of the polycondensation process; thus, the reaction cannot be driven further even with the presence of catalyst. The viscosity of PF1, PFII and PFIII samples were 83 cP, 117 cP and 165 cP, respectively. From these results, it can be deduced that the changes in molar ratio of formaldehyde did not significantly affect the viscosity of the produced resin synthesized at low temperature, via 40 °C. 

The viscosity values of the three PF resins prepared with different formaldehyde molar ratio and synthesized at 60 °C for 240 min were quite different. To illustrate, the final viscosity measured for these samples were 856 cP (PFI), 1195 cP (PFII) and 2305 cP (PFIII), respectively. For sample PFIII, the final measurement was only obtained up to 165 min because the sample was already solidified. The reaction was initiated at a much faster rate as indicated by the inflection point (on plot 2, 6 and 10) in Figure 4. The reaction was initiated approximately 50 min earlier when the molar ratio of formaldehyde was increased by 12%. Furthermore, with an increase in formaldehyde molar content, the bridging bonds between hydroxymethyl phenol and benzene rings were also increased causing the reaction rate to increase [27] and thus produce a more viscous product. The increase in molar content of formaldehyde also significantly affects the properties and the molecular structure of the produced PF resin which is evidently shown by the viscosity values presented in Figure 4.

At 80 °C, the data obtained from the different molar ratios of P to F samples provide some insights on the opportunity to control the reaction. From Figure 4, i.e., plots 3, 7 and 11, we can notice the different patterns displayed by the three plots. Even though all of the viscosity plots stopped at 3000 cP due to equipment limitations, the main difference between the three of them is the point of inflection when the reaction rate seemed to rapidly increase. For instance, for sample PFI, the inflection point when the reaction was rapidly increased, i.e., an increase of rate from 6.5 cP/min to a rate of about 142 cP/min, occurred at synthesis time of 120 min; whilst for PFII and PFIII, the rate of viscosity change occurred from 19 cP/min to 110 cP/min at t = 40 min and 43 cP/min to 149 cP/min at t = 15 min, respectively. The initial time recorded for all samples for this analysis was at 5 min after starting the experiment. These data show that as the molar content of formaldehyde was increased in the mixture, the time taken for the reaction to be driven from depolymerization towards polycondensation was shorter. The time taken for a complete solidification or curing of the product was approximately 10 times faster when the formaldehyde molar ratio was increased by 0.5. This scenario might also suggest that without a presence of cooling arrangement, i.e., heat was not removed efficiently from the system, an excessive heat, resulting from the heat generated by the reaction combined with the external heat supplied by the heater, might be accumulated within the system. The initial heat supplied by the heater was used to break down the polymer bond; therefore, higher molar ratio of formaldehyde provided more formaldehyde monomers that attached to the benzene rings during the addition stage and released more heat due to the exothermic reaction that occurred. The combined heat pushed the reaction further to the polycondensation stage where rigid complex polymers were formed via cross linking and water was removed via evaporation. 

Samples synthesized at 100 °C show similar viscosity profiles as those portrayed by samples synthesized at 80 °C. Rapid change in viscosity was observed as the molar ratio of formaldehyde to phenol was increased. Sample PFIII shows an instantaneous change of viscosity as illustrated by plot 12 (Figure 4c). The reaction was hardly controlled; some samples were exploded possibly due to extremely excessive heat generated and accumulated which led to the occurrence of thermal runaway reaction. At 100 °C, water was removed at a very fast rate making the solution highly viscous and volatile. Fast removal of water at 100 °C was evident based on the presence of water bubble within the cured sample as shown in Figure 5. From Figure 5, it can be inferred that the curing process occurred at a much faster rate than the liberation of water molecule because the water bubbles were still present on the surface of the cured resin. The bridging process of the monomers were much faster than the water evaporation process. This scenario led to the formation of defective laminate materials. Other works such as those presented by Park et al. [38], Żihorska-gotfryd [39] and Pizzi and Ibeh [40] illustrated that they were able to synthesize and produce good resins at 100 °C. This is hypothetically possible if they had installed a very good cooling system around the reactor. This cooling system was able to efficiently remove the heat generated by the reaction; however, this was not the case for our experimental set up. Table 2 shows the extracted data from the experiment.

The highlighted data (marked with *) represent the synthesis time when the recommended viscosity of 400–600 cP was achieved for a specific sample. Here, both the molar ratio of P to F and the synthesis temperature were defined. With the given information that relates time and viscosity, it is possible and much easier to control the quality of the PF resin produced. Zhao et al. [34] found that an increase in temperature shortened the synthesis time when they synthesized epoxy resin at a temperature between 50 to 80 °C. The slope of the viscosity vs time curve increases as the synthesis temperature was increased indicating an increase in reaction rate. The finding from this work is in line with that described by Zhao et al. [34]. 

### 3.2. Viscoelasticity of Phenol Formaldehyde (PF) Resin

The storage modulus, G′, loss modulus, G″ and tan δ of the phenol formaldehyde (PF) resin were evaluated for the synthesized PF resin. The viscoelasticity data provide information on the pliability of resins on the fiber. Figure 9 and Figure 10 show the graph of storage modulus, G′, graph of loss modulus, G″ and graph of tan δ per unit time (min). 

The changes of G′, G″ and tan δ of the PF resins prepared at different molar ratio of P to F (PFI, PFII, and PFIII) at synthesis temperature of 40, 60, 80 and 100 °C are shown in Figure 9 and Figure 10. The G′ plot represents the elastic character and reflects the solid-state behavior of the PF resin. The G″ plot portrays the viscous part and reflects the liquid-state behavior of the PF resin samples. Tan δ which is also known as the damping or loss factor, displays the ratio of G″ to G′. All samples measured were in liquid states; those that were already hardened were not included. Inset graphs were provided for Figure 9d and Figure 10d to provide more information of the fluids’ behaviors at the short observation period. In general, the G′ values are consistently higher than those of the G″ which indicate that the behavior of PF resin was gel-like [41,42] and that the material was highly structured; at some points, cross-linking process might also happen as that described by Moubarik [43].

At 40 °C, as illustrated by Figure 9a, the G′ values of all PF resins samples were increased in between the synthesis period of 60 to 120 min, which indicates that the viscosity of the liquid decreased during that period. Then, after 120 min the G′ values were decreased indicating that the PF resins are becoming more elastics, i.e., the viscosity increased. The viscosity of the PF resin synthesized at 40 °C was low at the beginning of the synthesis period (60 min to 90 min) most probably because of the i) depolymerization process of formaldehyde into monomers and ii) slow occurrence of addition reaction between phenol and formaldehyde monomers to produce cross-linked polymers and iii) absence of cross-linking process [20]. The deformation point for PF II and PF III occurs at 90 min; whilst for PFI, the deformation point was a bit later at 120 min. The retardation period for PFII and PFIII occurs at 180 and 120 min, respectively. Both PF II and PF III samples did not recover to their initial states possibly due to the reactions that had taken place enhanced with chemical bonding and cross-linking processes. PFI reached its recovery state at 240 min. This can be attributed to a very slow reaction that occurred during the observation period. If the reaction period was prolonged, a different viscoelastic behavior might be observed for sample PF I. 

Comparatively, graphs in Figure 9b–d do not show an apparent inflection point of G′ values for all PF resins samples synthesized at 60, 80 and 100 °C. The G′ and G″ values for PF I shown in Figure 9a–d are much higher compared to those of PF II and PF III. These can be attributed to the less viscous behaviors of the PFI samples as compared to those of PF II and PF III. When the molar ratio of formaldehyde and the synthesis temperature were increased, the G′ and G″ values were proportionally decreased. The G″ reached a constant value, i.e., reaching a plateau, indicating that the fluid was shifting from the liquid to the solid region. The large availability of formaldehyde monomers at higher formaldehyde molar ratio (1.00:1.75) escalated the formation of prepolymer cross linking and contributed to the fast polymerization of PF resin samples. For the purpose of this work, getting a cured PF resin is undesirable. Therefore, it is very important for the process to be controlled so that the plateau is not obtained. Figure 10 illustrates the viscoelastic behavior of all the PF resins samples synthesized in this work. 

The tan δ values presented provide some insights on the damping behavior of the produced PF resins. Based on Figure 10, the tan δ values for all PF resin samples are close to 0, indicating that they are showing ideal elastic behaviors in particular samples PF II and PF III that were synthesized at 60, 80 and 100 °C. According to Barnes et al. [32], when the tan δ value is about 0, the resin shows ideally elastic behavior, and when tan δ ≈ 1, the resin shows ideally viscous behavior. The trend shown by tan δ can also be related to the cross linking of polymers in the resin [44]. Here, the less viscous resin i.e., tan δ ≈ 1 shows low crosslinking possibly due to the absence or low concentration of prepolymers in the resin system and vice versa. The transition of tan δ value from high to low, i.e., approaching zero, indicates the transition of viscosity of the PF resin samples from less viscous to high viscous or possibly towards solid-gel state. Plateau regions, where tan δ ≈ 0, as shown in Figure 10b–d denote that the PF resins were cured much faster when the parameters such as the molar ratio of formaldehyde and the synthesis temperature were increased. Theoretically, it takes only nine min for Sample PF III to be cured when synthesized at 100 °C, compared to the same sample that was synthesized at 80 °C, in which it took about 20 min for the sample to be cured. However, since the intent of this work is to find the best suitable condition to prepare PF resin that can laminate, fully cured PF resin samples will be considered as defect samples. Therefore, the suitable samples will be those with a tan δ value between 10^−3^–10^−2^. For a sample with tan δ value that is higher than this, the sample will be very thin; whilst that with lower than this value, the sample is cured and can no longer laminate the desired surface.

## 4. Conclusions

The rheological behavior and viscoelasticity of phenol formaldehyde (PF) resin prepared using paraformaldehyde was assessed. The effects of the synthesis temperature and molar ratio of phenol to formaldehyde on the rheological behavior and viscoelasticity of the PF resins were investigated. The PF resin samples show non-Newtonian fluid behaviors. The rheology of the PF resin behaved as shear thinning when synthesized at low temperature (~40 °C) and as shear thickening when synthesized at much higher temperature, i.e., more than 60 °C. The storage modulus (G′), loss modulus (G″) and tan δ of PF resin were also influenced by the synthesis temperature. The effect of molar ratio, i.e., formaldehyde content, was not very significant when the synthesis was performed at 40 °C possibly due to the slow reaction that happened in the system. At much higher synthesis temperature, e.g., 100 °C, the formaldehyde content significantly affects the rate of reaction of the system. The drastic viscosity changes with the heat explosion that occurred indicates that the reaction was highly exothermic and was uncontrolled when the cooling arrangement was absent. This work proposes suitable operating conditions, i.e., synthesis time, when the parameters, e.g., synthesis temperature and the molar ratio of P to F, are known.

## Figures and Tables

**Figure 1 materials-13-02578-f001:**
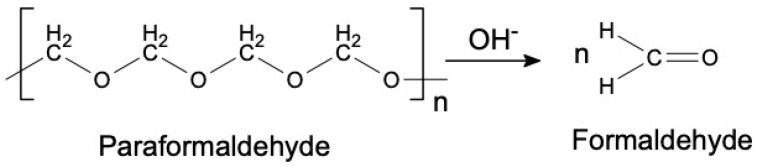
Depolymerization of paraformaldehyde [25].

**Figure 2 materials-13-02578-f002:**
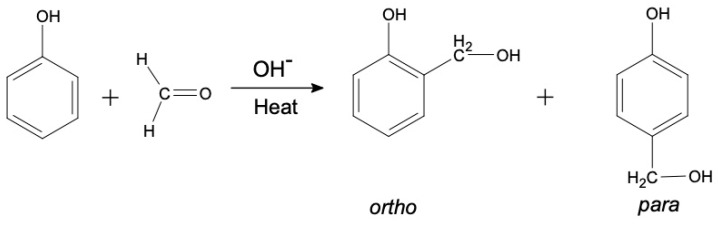
Addition reaction of formaldehyde to phenol [12].

**Figure 3 materials-13-02578-f003:**
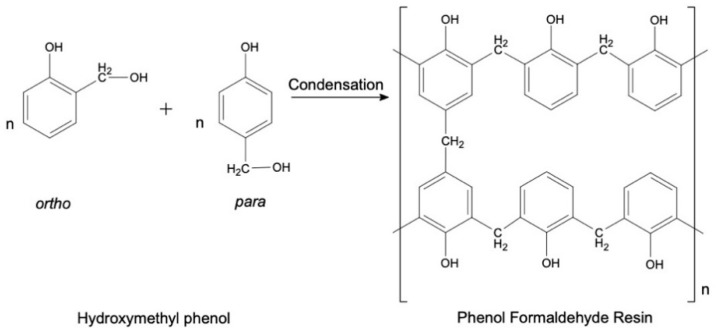
Condensation reaction of hydroxymethyl phenol [12].

**Figure 4 materials-13-02578-f004:**
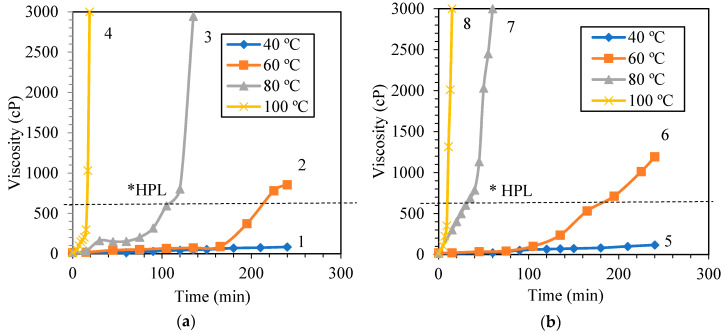
Effect of synthesis temperature on the dynamic viscosity of PF resins prepared at different molar ratios of P to F (**a**) PF I, (**b**) PF II and (**c**) PF III. * Dashed line refer to high pressure laminate (HPL) standard requirement for viscosity [22].

**Figure 5 materials-13-02578-f005:**
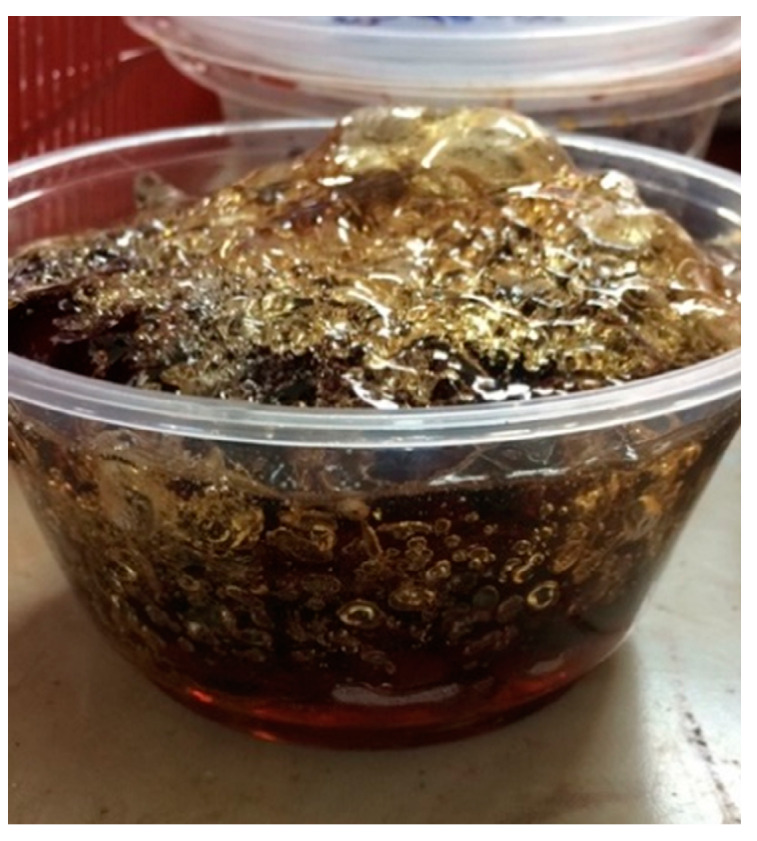
Sample of phenol formaldehyde (PF) resin prepared at 100 °C.

**Figure 6 materials-13-02578-f006:**
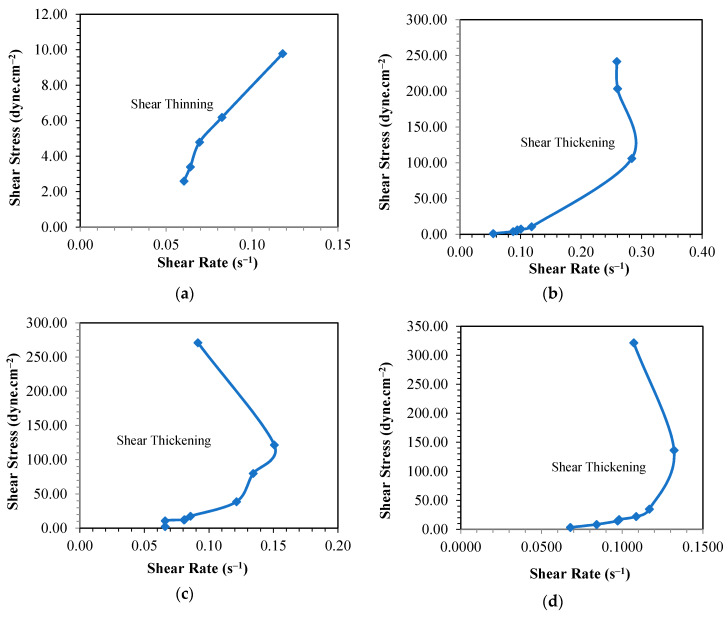
Curve of shear rate—shear stress for phenol formaldehyde (PF) resin (PF I) synthesis temperature at (**a**) 40, (**b**) 60, (**c**) 80 and (**d**) 100 °C.

**Figure 7 materials-13-02578-f007:**
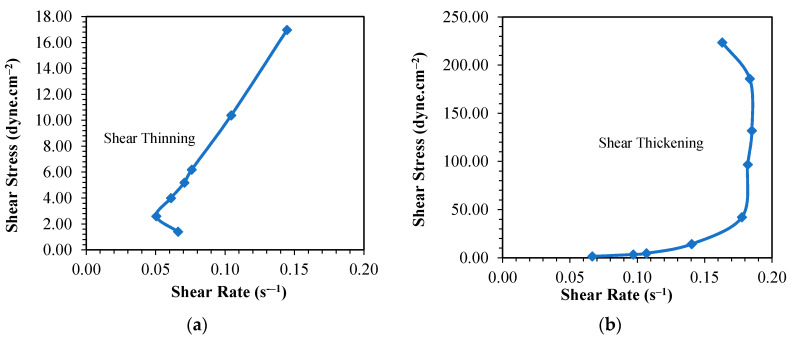
Curve of shear rate—shear stress for phenol formaldehyde (PF) resin (PF II) synthesis temperature at (**a**) 40, (**b**) 60, (**c**) 80 and (**d**) 100 °C.

**Figure 8 materials-13-02578-f008:**
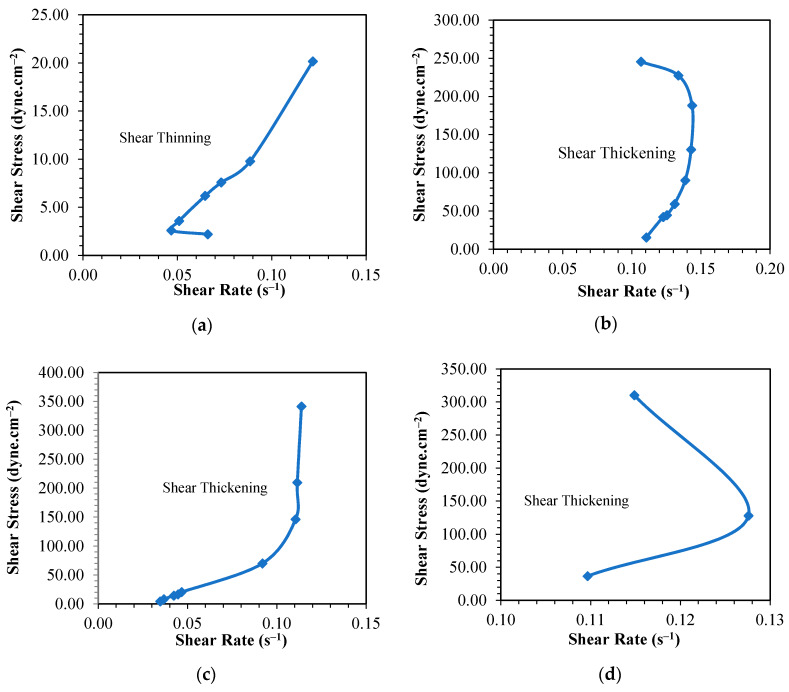
Curve of shear rate—shear stress for phenol formaldehyde (PF) resin (PF III) synthesis Temperature at (**a**) 40, (**b**) 60, (**c**) 80 and (**d**) 100 °C.

**Figure 9 materials-13-02578-f009:**
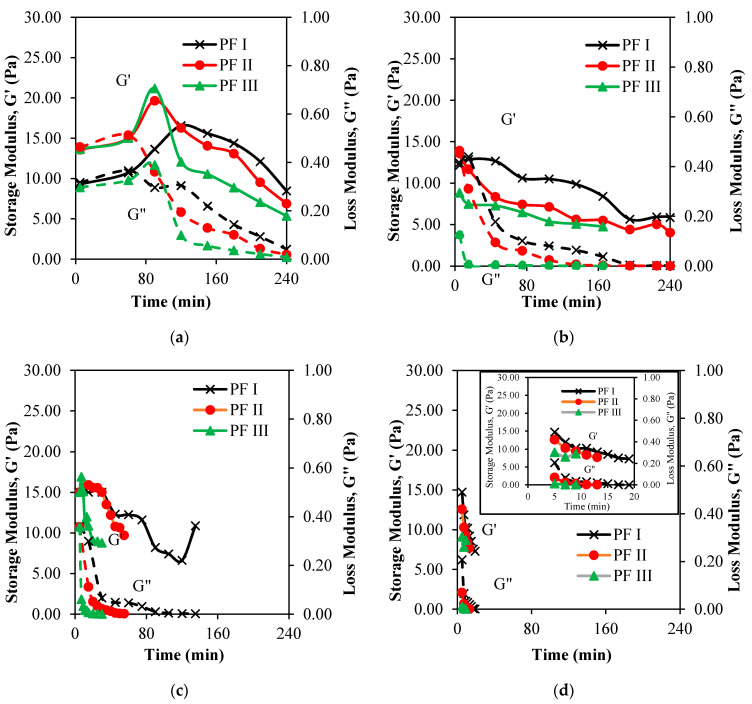
Graphs of storage modulus, G′—loss modulus, G″ per unit time of phenol formaldehyde samples (PF I, PF II and PF III) synthesized at (**a**) 40; (**b**) 60; (**c**) 80; (**d**) 100 °C.

**Figure 10 materials-13-02578-f010:**
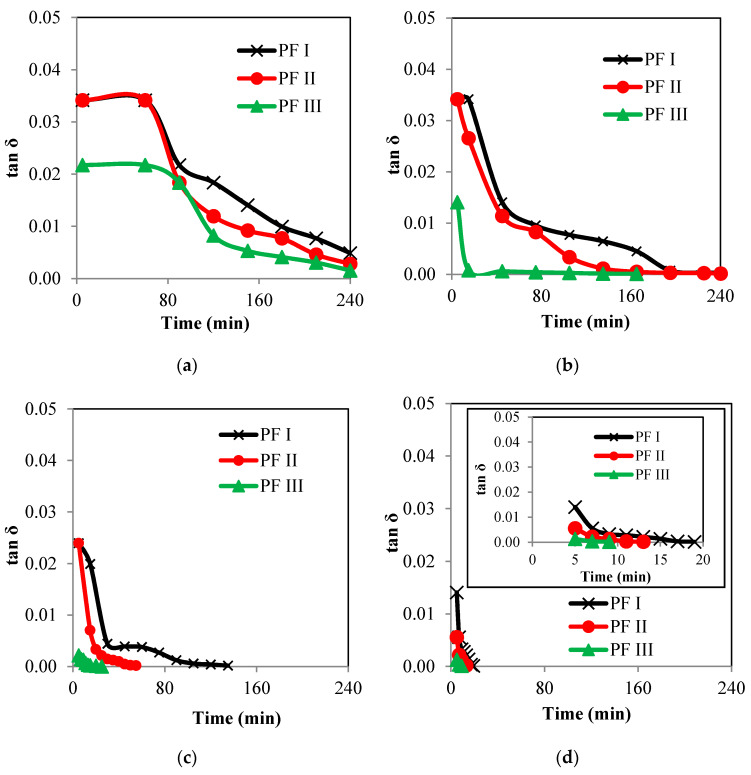
Graphs of tan δ per unit time of phenol formaldehyde samples (PF I, PF II and PF III) synthesized at (**a**) 40; (**b**) 60; (**c**) 80; (**d**) 100 °C.

**Table 1 materials-13-02578-t001:** Set conditions for the synthesis of phenol formaldehyde (PF) resin.

Sample	Molar Ratio of Phenol to Formaldehyde (P:F)	Synthesis Temperature (°C)	Synthesis Period (min)	Sampling Intervals (min)
PF I	1.00:1.25	40	240	30
	1.00:1.25	60	240	30
	1.00:1.25	80	240	15
	1.00:1.25	100	240	2
PF II	1.00:1.50	40	240	30
	1.00:1.50	60	240	30
	1.00:1.50	80	240	5
	1.00:1.50	100	240	2
PF III	1.00:1.75	40	240	30
	1.00:1.75	60	240	30
	1.00:1.75	80	240	2
	1.00:1.75	100	240	2

**Table 2 materials-13-02578-t002:** Synthesis time with the viscosity of PF resin.

Molar Ratio of Phenol to Formaldehyde (PF) Resin	Synthesis Temperature (°C)	Synthesis Time (min)	PF Resin Viscosity (cP)	Remarks
PF I (1.00:1.25)	40	240	83	Runny
	60	195	373 *	Slightly viscous
		240	850	Very viscous
	80	105	595 *	Viscous
		135	2946	Almost Solid
	100	15	299	Runny
		20	3000	Solid
PF II (1.00:1.50)	40	240	117	Runny
	60	165	532 *	Viscous
		240	1192	Highly viscous
	80	30	492 *	Viscous
		60	3000	Solid
	100	9	351 *	Slightly viscous
		15	3000	Solid
PF III (1.00:1.75)	40	240	165	Runny
	60	15	450 *	Viscous
		165	2305	Highly viscous
	80	13	436 *	Viscous
		30	3000	Solid
	100	5	335 *	Slightly viscous
		12	3000	Solid

* Viscosity in which the sample is within the pliable range.

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
