# Peer review of "Rheological Study of Phenol Formaldehyde Resole Resin Synthesized for Laminate Application"

_materials, 2020, doi:10.3390/ma13112578_

Round 1
Reviewer 1 Report
The authors present an interesting study on how changing conditions for the reaction between phenol and paraformaldehyde (formaldehyde) impacts rheological characteristics. However, the manuscript is full of grammatical errors and awkward wording/phrasing that make many sections extremely difficult to follow. In addition, there are some serious flaws with the experimental design that make it impossible to make true conclusions. Given the high impact factor of Materials, I must reject this manuscript.
General Comments:
- A reaction diagram (Lines 62-65) would provide clarity.
- Line 115 “set at +/- 150.00 rpm”?
- Figure 1: Not clear why “* HPL” is on every graph and what message is being portrayed. Also, I have no idea what the different letters are referring to. If these are stats, the stats tests used and confidence intervals are nowhere to be found in the Figure, or in the Materials and Methods. Also, if these are stats, what is being compared? The letters just seem to be placed at random points along the line, especially given that so many measurements were actually taken. There are also no values above 3000 cP, so why does the graph extend to 4500 cP? This just makes the data harder to see.
- Lines 151: Doesn’t a “0 cP/minutes” reaction rate indicate no reaction?
- Table 2: All of the discussion in section 3.1.1. discusses the impact of synthesis temperature, but from what I can see in Table 2, the synthesis time was also varied. Scientifically speaking, they are not comparing apples to apples, and thus I think that they are overinterpreting their data. I understand that they are making the claim that at 100 degrees, curing occurred which prevented longer synthesis times, and that at 40 degrees, there was virtually no reaction even with extended times, but it is unclear why you would change the synthesis times within a given system (i.e. PH I) or between the different systems (i.e. 80 degrees in PF1 and PFII). In the latter example, the synthesis times are 105 and 30 minutes, respectively, so how can the two systems really be compared at all?
- Watch for repetition. For example, the discussion around the 100 degree C samples hardening and not being used for further experiments is mentioned several times and in different sections. Interestingly, after mentioning this, reactions from 40 and 100 degrees are still characterized (i.e. shear rate, viscoelasticity). In another more obvious case, The paragraph on Lines 187-191 are essentially cut and pasted in Lines 212-215.
- The explanation for choosing the synthesis time (lines 219-229) is not clear. Added to this, I am not sure how a study on how reaction conditions impact viscosity of a resin (the primary objective of this paper) can be achieved if more than one variable is changed at a given time, and for different reasons. THE RESEARCH METHODOLOGY EMPLOYED SUFFERS FROM CONFORMATION BIAS.
- Discussion is lacking on the three “Curve of Shear Rate”. What conclusions can be drawn about the reaction, and how does this impact the ability for this resin to be used as a laminate? There is no discussion about which set of conditions was most promising.
- Overall, the impact is a little lost on me as the main conclusions are that the reaction proceeds faster with temperature and time, which is something that is already well-known. The novelty of this work is unclear.
Reviewer 2 Report
Revision materials-736365: Rheological Study of Phenol Formaldehyde Resole Resin Synthesized for Laminate Application.
The manuscript assesses the rheological behavior of PF resin using paraformaldehyde during synthesis for laminate application, which is of scientific and industrial interest. Few publications deal with heat explosions and runaway reactions, therefore the problem is significant and concisely stated. The abstract is concise, however, the language and the grammar should be revised and then this paper would deserve publication in Materials.
Comments for authors:
Heat explosions and runaway reactions are mentioned in the abstract and in the introduction, but no conclusions are drawn at which conditions they may occur and how to avoid them. Please address this issue. Generally, please revise the use of present and past tense as well as the grammar throughout the whole paper.
- 3 line 100: Either “investigate” or “study”
- 3 line 108 should read: ... The rheological properties of the phenol formaldehyde produced in this work 108 were investigated at different temperatures, times and ratios...
- 4 lines 124-125: should be Figure 1
- 5 line 160 should read: ... This was shown by...
- 6 lines 169-170 should read: ... indicates that the reaction is the temperature dependent...
- 6 line 170 should read: ... that an increase in temperature shortens the...
- 6 line 174 should read: ...increase of resin cross-linking...
- 6 line 177 + p. 7 line 208 + p. should read: ... resin samples...
- 6 line 187 + p. 8 line 212 + p. should read: ... resin sample...
- 7 line 205 should read: ... due to the increase of resin cross-linking...
- 8 line 211 should read: ...1(a), (b) and (c).
- 8 line 217 should read: ...The synthesis time was chosen...
- 8 lines 223-224: I don´t understand this sentence, please rephrase.
- 8 line 226 + p. 8 line 227 should read: ... will occur
- 8 line 228 should read: ... was chosen due to ...
- Table 2: For PF I, the third entry “30 min, 60 ºC” appears to be redundant
- 9 lines 256-257 should be similar to as follows: ... The curves show the non-Newtonian behavior of the resins....
- 11 line 271 should read: ...This is probably due to...
- 11 line 273 should read: ... will decrease the gel time/....
- 11 line 276 should read: ... occur...
- 11 lines 274-276 should read: ...most probably because only the addition reaction of formaldehyde to phenol occurred and no polymerization and condensation reaction (formation of methylene and ether bridges) occurred at...
- 12 lines 294 ff. should read: Figure 6 (a) shows that G’ for all sets of PF resins increased in the range of synthesis time 60 minutes to 120 minutes, which indicates that the viscosity of liquid PF decreased. Then, the G’ decreased as the synthesis time increasd to 240 minutes showing that the viscosity of PF resins was increased at 40 ℃ synthesis temperature. The viscosity of the PF resin decreased for synthesis temperature at 40 ℃ most probably because of the slow reaction of phenol and formaldehyde at low synthesis temperature [20]. Figure 6 (b), (c) and (d) show that G’ for all sets of PF resin decreased with the increase of synthesis time which indicates that the viscosity of the PF resins increased. For all sets of PF resins and synthesis temperature, the G’ values were higher than G” which indicates that the behavior of PF resin was gel-like [38,39] and that the material was highly structured. Higher G’ and G” values for PF I for all sets of synthesis temperature Figure 6 (a) to (d) indicate the less viscous behavior of the liquid resin as compared to PF II and PF III, respectively.
- 13 lines 308-309 should read: Figure 7 shows that PF I gave the highest value of tan δ in all operating temperature conditions. It shows that PF III has elastic behavior followed by PF II and PF I, respectively.
- 13 lines 311: ... also can be related to...
- 14 line 325: ..., G’ increased with...
- 14 line 326: ..., G’ increased with...
- 14 line 330: ...All authors have red...
Reviewer 3 Report
Comments and Suggestions for the Authors
In the article presented for review, the Authors evaluated the rheological properties of phenol formaldehyde resin synthesized at different ratios of raw materials (phenol and paraformaldehyde), at different operating temperature and time. Comments and suggestions to Authors that may improve readability of manuscript:
1. The results described in the manuscript are presented in four tables and seven figures. The figures show the following relationships:
- viscosity vs time (Figure 1),
- flow curves (shear stress vs shear rate) (Figures 3, 4, 5),
- storage modulus and loss modulus vs time (Figure 6),
- tan delta vs time (Figure 7).
In the "Materials and methods" chapter, the authors gave a method for the viscosity measurement. The Authors wrote that the viscosity measurements were carried out using the Brookfield viscometer at 200 rpm, in the temperature range of 25-40 ° C. Because of the rheological properties strongly depend on temperature, the authors should clearly indicate at which specific temperature the measurements were made. This applies to the data given in figures 1, 3, 4 and 5 and the values ​​given in table 2).
2. The Authors did not specify how they determined the relationships presented in Figures 3, 4, 5. Nor did the Authors describe the method for measuring viscoelastic properties.
3. What is the purpose of placing "* HPL" in Fig. 1 (a), (b), (c)? Was this to indicate a viscosity level of ~ 500 cP (page 6, line 176)? In the current presentation it is not legible and seems unnecessary.
4. It seems that after the text on page 5, lines 149-152:
”… The synthesis temperature influenced the viscosity plots as in Figure 1. Based on Figure 1 (a) to 149 (c), all plots for synthesis temperature of 40 ℃ had very small gradient, i.e. slope a, h and n of the 150 curve is ~ 0 cP/minutes indicates the reaction rate between phenol and paraformaldehyde are at its 151 slowest.”
there should be a reference to table 3.
5. On page 6 (line 177) the Authors wrote: "...Based on this, all PF resins samples that have viscosity higher than 500 cP were eliminated."
On the same page below (lines 188-190) there is the text: "... Only samples prepared at 60 ℃ and 80 ℃ formed fluid with acceptable viscosity range (± 100 cP from set value) that can form layers during the laminating process."
The content of both sentences seems to be contradictory (500 cP + 100cP = 600 cP; 600 cP > 500 cP).
6. The text on pages 7 and 8 (lines 205-213) is very similar to the text on pages 6 (lines 178-188). Is it necessary to repeat this text?
7. On page 8 (line 222), the Authors refer the reader to table 5. Table 5 is not included in the manuscript.
8. The text on page 12 (lines 295-298) is not clear:
"... Then, the G’ decrease as the synthesis time increase to 240 minutes showing that the viscosity of PF resins is increase at 40 ℃ synthesis temperature. Decreased in the PF resin viscosity for synthesis temperature at 40 ℃ most probably because of the slow reaction of phenol and formaldehyde at low synthesis temperature [20] …"

Round 2
Reviewer 1 Report
I appreciate the authors' efforts with their significant revisions, which have helped to clarify and alleviate my major concerns. I believe that the paper does have scientific merit and the experiments have been conducted properly, which the authors' conclusions acceptable. My only concern would be regarding novelty and thus impact.